# Regulation of CREB Phosphorylation in Nucleus Accumbens after Relief Conditioning

**DOI:** 10.3390/cells10020238

**Published:** 2021-01-26

**Authors:** Elaheh Soleimanpour, Jorge R. Bergado Acosta, Peter Landgraf, Dana Mayer, Evelyn Dankert, Daniela C. Dieterich, Markus Fendt

**Affiliations:** 1Institute for Pharmacology and Toxicology, Medical Faculty, Otto-von-Guericke University Magdeburg, 39120 Magdeburg, Germany; bergado358@yahoo.com (J.R.B.A.); peter.landgraf@med.ovgu.de (P.L.); dana.mayer@med.ovgu.de (D.M.); evelyn.dankert@med.ovgu.de (E.D.); daniela.dieterich@med.ovgu.de (D.C.D.); markus.fendt@med.ovgu.de (M.F.); 2Center for Behavioral Brain Sciences, Otto-von-Guericke University Magdeburg, 39106 Magdeburg, Germany

**Keywords:** relief learning, nucleus accumbens, CREB, Western blot

## Abstract

Relief learning is the association of environmental cues with the cessation of aversive events. While there is increasing knowledge about the neural circuitry mediating relief learning, the respective molecular pathways are not known. Therefore, the aim of the present study was to examine different putative molecular pathways underlying relief learning. To this purpose, male rats were subjected either to relief conditioning or to a pseudo conditioning procedure. Forty-five minutes or 6 h after conditioning, samples of five different brain regions, namely the prefrontal cortex, nucleus accumbens (NAC), dorsal striatum, dorsal hippocampus, and amygdala, were collected. Using quantitative Western blots, the expression level of CREB, pCREB, ERK1/2, pERK1/2, CaMKIIα, MAP2K, PKA, pPKA, Akt, pAkt, DARPP-32, pDARPP-32, 14-3-3, and neuroligin2 were studied. Our analyses revealed that relief conditioned rats had higher CREB phosphorylation in NAC 6 h after conditioning than pseudo conditioned rats. The data further revealed that this CREB phosphorylation was mainly induced by dopamine D1 receptor-mediated activation of PKA, however, other kinases, downstream of the NMDA receptor, may also contribute. Taken together, the present study suggests that CREB phosphorylation, induced by a combination of different molecular pathways downstream of dopamine D1 and NMDA receptors, is essential for the acquisition and consolidation of relief learning.

## 1. Introduction

Relief learning is the association of a cue with the relief from an aversive event and can be observed across species, e.g., in humans, rodents, or flies [1]. Since the relief of an aversive event is rewarding [2,3], the associated cue—also called conditioned relief stimulus—later induces appetitive behaviors such as approach response or attenuates defensive behavior like the startle response [1]. There is an increasing interest in understanding the mechanisms underlying relief learning since pathological changes of relief learning are discussed for human diseases such as non-suicidal self-injury, trichotillomania but also anxiety disorders [1,4]. Furthermore, pain relief learning is used to investigate the affective/motivational aspects of chronic pain and its treatment [5]. Studies in humans and rodents identified the nucleus accumbens (NAC) and the mesolimbic dopamine system as critical brain structures for relief learning [2,6,7]. Similar to appetitive instrumental learning [8], relief learning is mediated by a coincident activation of NMDA and dopamine D1 receptors within the NAC [9,10].

While there is some basic knowledge on the neuroanatomical and neuropharmacological basis of relief learning in mammals [1,7,10] the molecular basis of it has been poorly investigated so far. Hence, the aim of the present study was to explore how relief learning affects different signaling pathways within the NAC and other brain areas being associated with relief learning. Due to the crucial role of accumbal NMDA and dopamine D1 receptors in the acquisition of conditioned relief [10], the signaling pathways associated with these two receptors were in the focus of the present study.

A key protein in synaptic plasticity and memory is the cAMP response element-binding protein (CREB) which functions as a transcriptional activator after its phosphorylation [11]. Previous studies in hippocampus-dependent contextual fear learning showed a biphasic pattern for CREB phosphorylation in long-term memory formation [12,13,14]. In the first phase (within 1 h after conditioning) CREB phosphorylation is likely to be stress-related and can trigger transcription of immediate early genes. In the second phase (between 3 and 6 h after conditioning), CREB phosphorylation can modulate the expression of downstream target genes that are related to memory consolidation [14].

Based on these findings, protein expression and phosphorylation patterns after relief conditioning in male rats were analyzed at two time points, 45 min, and 6 h, in the present study. First, CREB expression and phosphorylation levels were examined in the NAC and four other brain regions that at least partly may play a role in relief learning (prefrontal cortex, hippocampus, amygdala, dorsal striatum). Second, to identify the molecular pathways related to CREB phosphorylation in the nucleus accumbens during relief conditioning, the expression of selected kinases in different signaling pathways underlying NMDA and dopamine D1 receptor activation was investigated. 

## 2. Materials and Methods

### 2.1. Animal

Male *Sprague-Dawley* rats, 7 to 9 weeks old at the time of conditioning were used. They were bred in our animal facility (original breeding stock: Taconic, Silkeborg, Denmark) and kept in temperature-controlled rooms (22 ± 2 °C; 50 ± 10% humidity) under a light/dark cycle of 12/12 h (lights on at 6:00 a.m.) with food and water ad libitum. Experimental subjects were housed in groups of 4 to 6 in standard cages filled with wood chip bedding (Macrolon type IV; floor area: 1820 cm^2^; Tecniplast, Hohenpeißenberg, Germany). Behavioral tests and tissue isolation were performed during the light phase from 9:00 A.M. to 4:00 P.M. All studies were conducted under the European regulations for animal experiments (2010/63/EU) and approved by the local authorities (Landesverwaltungsamt Sachsen-Anhalt: Az. 42502-2-1172 and 42502-2-1309 Uni MD).

### 2.2. Apparatus

Behavioral experiments were performed in a startle system consisting of wooden chambers (35 × 35 cm and 38 cm high; SR-LAB, San Diego Instruments, San Diego, CA, USA) equipped with animal cylindric enclosures (9 cm inner diameter and 16 cm in length) made of acrylic glass. Motion-sensitive transducers mounted underneath these enclosures detected the movements of the animals. For data acquisition, the output signal of the transducers was digitized at a sampling rate of 1 kHz and stored on a computer. As startle magnitudes, the mean transducer output in the time window 10 to 30 ms after the startle probe onset was taken [9,10].

As a conditioned stimulus (CS), a light cue (5 s; ~1000 lux) was presented to the animals with a 10 W white bulb mounted on the back of the test chambers. As an unconditioned aversive stimulus, scrambled foot shocks (0.5 s; 0.4 mA) were administered by a floor grid of six parallel bars (0.5 cm in diameter and 1 cm apart) inserted into the enclosures during conditioning. A white background noise of 50 dB sound pressure level (SPL) and the acoustic startle probe (40 ms; 96 dB SPL white noise) were generated by a loudspeaker mounted at the ceiling of the chambers.

### 2.3. Behavioral Procedure

On the first day, a baseline startle test was performed. The rats were introduced into the enclosures of the startle boxes and after an acclimation time of 5 min, 10 startle probes stimuli with an inter-trial interval (ITI) of 30 s were presented. On day 2, animals were conditioned. For relief conditioning, the US, followed by the CS (US offset to CS onset: 2.5 s), were presented 15 times with a mean ITI of 150 s. In the control-conditioning group, the US and the CS presentations were uncorrelated in time (i.e., randomly presented). No startle probes were presented during the conditioning session. Forty-five minutes or 6 h after the conditioning procedure, most of the rats were sacrificed for tissue collection (see below). However, a small sample of animals was subjected to a retention test one day later. The retention test consisted of 5 min of accommodation time, followed by 10 startle probes to habituate the startle response. Then, 20 further startle stimuli were presented, half of them in the absence and half of them in the presence of the light CS (in a pseudorandomized order; ITI: 30 s).

### 2.4. Experimental Design

The baseline startle test was used to group the rats into the groups “Relief” and “Control” with similar mean baseline startle magnitudes. One day later, as shown in Figure 1, they were either relief or control conditioned. A further group of rats stayed naive, i.e., they remained in the home cage for both days. In all groups, rats were randomly assigned to the 45 min or 6 h time points. At these time points, brain samples were taken (in naive rats at the same time of the day as conditioned subjects, to control for circadian modulation in the expression of the targeted proteins). Some animals in the Relief and Control group were not used for brain dissection but were tested on day 3 in a retention test.

### 2.5. Tissue Isolation

Rats were sacrificed 45 min or 6 h after the conditioning session. They were capitated under anesthesia with isoflurane (Baxter Germany GmbH; Unterschleißheim, Germany) and the brains were quickly extracted and dissected in coronal slices (1 mm) using rat brain slicer matrix (PA001, David Kopf Instruments, Tujunga, CA, USA). The brains were dissected on ice into 5 different regions, namely the prefrontal cortex (PFC), nucleus accumbens (NAC), dorsal striatum (STR), dorsal hippocampus (HIP), and amygdala (AMY) [15]. Samples were immediately deep frozen after dissection and stored at −80° C until used for protein extraction followed by Western blot analysis.

### 2.6. Protein Extraction (from Brain Samples) and Western Blot Analysis

The brain samples were homogenized on ice in 1xPBS (pH 7.4), containing proteinase inhibitor (cOmplete™ Protease Inhibitor Cocktail, Roche, Basel, Switzerland), phosphatase inhibitor (PhosSTOP™, Roche, Basel, Switzerland), and 250 U Benzonase^®^Nuclease (Sigma-Aldrich, St.Louis, MI, USA # SLBS8402) using K-Control™ TLC homogenizer. For protein extraction the homogenized samples were solubilized in 4xSDS sample buffer containing 1% (*w*/*v*) SDS, 40% (*v*/*v*) glycerol, 20% (*v*/*v*) β-mercaptoethanol, 250 mM Tris pH 6.8, 0.004% (*w*/*v*) Bromophenol blue and, boiled at 95 °C for 5 min. Protein quantification was performed by amido black assay [16]. For Western blot analysis 18 µg protein per lane was loaded on a 5–20% gradient SDS-PAGE, separated with a constant current of 10 mA, and subsequently transferred on a nitrocellulose membrane (LI-COR, Lincoln, NE, USA, # 926-31092). The membranes were incubated with blocking buffer (5% non-fat dried milk or 5% BSA-Albumin fraction V, ROTH #8076.5- in 1xTBS, 0.1% Tween, depending on the antibody) for 60 min at room temperature and finally incubated with the following primary antibodies; CREB (Cell signaling, Denver, MA, USA, #919, 1:1000), Phospho-CREB (Ser133) (Cell Signaling, Denver, MA, USA, #9198, 1:1000), ERK1/ERK2 (R&D system, Minneapolis, MN, USA, #MAB1576, 1:1000), Phospho-ERK1 (T202, Y204)/phospho-ERK2 (T185/Y187) (R&D system, Minneapolis, MN, USA, #AF1018, 1:2000), Akt (pan) (Cell Signaling, Denver, MA, USA, #2920, 1:2000), Phospho-Akt (Thr308) (Cell Signaling, Denver, MA, USA, #13038, 1:1000), PKA (Cell Signaling, Denver, MA, USA, #4782, 1:1000), Phospho-PKA (Thr197) (Cell Signaling, Denver, MA, USA, #4781, 1:1000) CaMKII alpha (Abcam, Cambridge, UK, #ab92332, 1:1000), 14-3-3 (Abcam, Cambridge, UK, #ab9063, 1:1000), Neurologin2 (SySy, Göttingen, Germany, #129511, 1:1000), MAP2K (Proteintech, Rosemont, IL, USA, #17340-1-AP, 1:1000), DARPP-32 (19A3) (Cell Signaling, Denver, MA, USA, #2306, 1:1000), Phospho-DARPP-32 (Thr34) (Cell Signaling, Denver, MA, USA, #12438, 1:1000), Beta-Actin (Cell Signaling, Denver, MA, USA #8457, 1:1000), and Beta-Actin (Cell Signaling, Denver, MA, USA, #3700, 1:2500) overnight at 4 degree. Following four times 5-min washing with 1xTBS and 1xTBST, the corresponding horseradish peroxidase-conjugated secondary antibodies (1:7500, Dianova, Hamburg, Germany) were added to the membrane for 90 min at room temperature. The blots were washed 4 times in PBS and PBST for 5 min each and treated with chemiluminescence substrate (ECL, Thermo Fischer, Waltham, MA, USA, #32106), West Dura Extended Duration Substrate (DURA, Thermo Fischer, Waltham, MA, USA, #34076), and West Femto Maximum Sensitivity Substrate (FEMTO, Thermo Fischer, Waltham, MA, USA, #34096), based on the protein-dependent signal intensity. The density of the bands was measured by Image Studio Lite version 5.2.5 and normalized to the ß-actin signal from each blot. Western blot was performed in triplicate for each sample. 

### 2.7. Statistical Analysis

Data were expressed as means ±SEMs. Behavioral data were analyzed with analysis of variance (ANOVA). Planned comparisons of protein expression and phosphorylation in control- and relief conditioned rats were performed with non-parametric t-tests (Prism 8.0, GraphPad Software Inc., La Jolla, CA, USA). A significance level of *p* ≤ 0.05 was used for all tests. Outliers (Q = 5%) were removed from the analysis.

## 3. Results

### 3.1. Behavioral Analysis

In the retention test on conditioned relief, the light stimulus attenuated the startle magnitude in the relief-conditioned rats but not in the control-conditioned rats (Figure 2). This is supported by an ANOVA using group (Control, Relief) as between-subject factor and trial type (startle alone, CS-startle) as within-group factor. There was an interaction between group and trial type (*F*_1,20_ = 7.31, *p* = 0.01) and a significant effect of trial type (*F*_1,20_ = 10.87, *p* = 0.004) but not of group (*F*_1,20_ = 1.28, *p* = 0.28). Furthermore, post-hoc Sidak’s tests showed significant startle attenuation in the Relief group (*t*_11_ = 4.24, *p* = 0.0008) but not in the Control group (*t*_11_ = 0.42, *p* = 0.90). 

### 3.2. Analysis of Protein Expression, Protein Phosphorylation, and Phosphorylation Ratios

Figure 3, Figure 4, Figure 5 and Figure 6 depict the levels of protein expression and protein phosphorylation as well as the phosphorylation ratio (calculated within each sample) of the control- and relief-conditioned rats that were normalized to the mean levels of the naive animals. Since we were interested in the molecular basis of relief learning, we focused our statistical analyses on the comparison of the control- and relief-conditioned animals, i.e., group Control vs. group Relief. Differences between these groups should be caused by associative learning during relief conditioning. However, of note, animals of both of these groups were exposed to the conditioning setup, light stimuli, and electric stimuli, whereas this was not the case in naive animals. This means that differences between the naive group and the control- and relief-conditioned groups may represent molecular changes induced by the exposure to set up, light, and electric stimuli.

#### 3.2.1. Relief Learning Affects CREB Expression Time-Dependent and Brain Region-Specifically 

The first part of the protein analysis was focused on CREB expression and phosphorylation, 45 min and 6 h after conditioning. Figure 3 depicts the mean of these parameters, normalized to the levels in the naive group, as well as the mean phosphorylation ratio (pCREB/CREB × 100).

In the NAC (Figure 3a), total CREB expression after relief conditioning was not different in the two groups after 45 min and 6 h. Of note, after 45 min the level of pCREB was attenuated in relief-conditioned rats compared to the control group (*p* = 0.04). After 6 h, pCREB (*p* = 0.007), as well as phosphorylation ratio of CREB (*p* = 0.04), were significantly higher in relief-conditioned animals compared to the control group.

In the HIP (Figure 3b), the expression level of total CREB and the pCREB were significantly decreased 45 min after relief conditioning *(p* = 0.003 and *p* = 0.03, respectively). After 6 h, no significant differences between the two conditioned groups were seen, but a clear tendency towards higher phosphorylation could be observed in the fraction of relief learners. Interestingly, in the PFC (Figure 3c), relief conditioning leads to a significant decrease of pCREB (*p* = 0.01) and the phosphorylation ratio (*p* = 0.002) after 6 h, while no differences were seen after 45 min. In STR and AMY, the expression and activation level of CREB in both conditioned groups, as well as both time points, were similar.

#### 3.2.2. Moderate Induction of Kinases Underlying NMDA Receptor Signaling Plays a Role in Relief Learning

We have previously shown, that accumbal NMDA receptor activation plays a key role in the acquisition of conditioned relief [9]. NMDA receptor activation can induce CREB phosphorylation via different signaling pathways. One of the pathways involves MAP2K that regulates MAPK kinase (MEK). MEK phosphorylation then can activate ERK that phosphorylates CREB at Ser-133 [17]. An alternative pathway for NMDA receptor mediated CREB phosphorylation includes the Ca^2+^-calmodulin-dependent kinase (CaMK) that is activated by the calcium influx [18]. To determine which of these signaling pathways and kinases mediates the CREB phosphorylation during relief learning, the expression levels ERK1/2, pERK1/2, CaMKIIα, and MAP2K were analyzed in the five selected brain regions at both time points.

In the NAC, 45 min after relief conditioning, the expression level of these candidate kinases was not affected, while after 6 h a moderately but not significantly higher level of pERK and CaMKIIα expression was observed in relief-conditioned rats (Figure 4a,b, Appendix A). However, there were significant group differences when pERK and CaMKIIα expression levels were analyzed together (Figure 4c; ANOVA, factor group: *F*_1,20_ = 6.20, *p* = 0.02,). This suggests that relief conditioning activates both pathways together and that CREB phosphorylation could be mediated by their joint activation. There were no effects of relief conditioning on these kinases in the four other brain regions at both time points (see Appendix A). 

#### 3.2.3. Dopamine D1 Receptor Mediated PKA Activation Is Involved in CREB Phosphorylation after Relief Conditioning

In addition to NMDA receptor activation, stimulation of dopamine D1 receptor is required for relief learning, too [10]. The cAMP-PKA signaling pathway is the main downstream effector of dopamine receptors. This pathway also leads to the phosphorylation of CREB at Ser-133 and plays an important role in learning and memory [9,10]. Therefore, we also measured the expression and phosphorylation level of kinases involved in dopamine D1 receptor-mediated signaling in the NAC (Figure 5). As depicted in Figure 5a, 6 h after conditioning in the relief-conditioned rats, the level of pPKA and the phosphorylation ratio of PKA (pPKA/PKA × 100) were significantly increased *(p* = 0.01 and *p* = 0.05, respectively) compared with the control-conditioned (Figure 5a,d). 

In addition to PKA, dopamine receptors can also modulate the activity of Akt (PKB). Akt phosphorylation can occur at Thr308 or Ser473. In this study, the Akt expression and phosphorylation on Thr308 were analyzed. After 45 min, no Western blot signal of pAkt was detectable in the NAC, and after 6 h, no effects of relief conditioning on Akt expression and phosphorylation were found (Figure 5b,e). 

Via PKA and Ca^2+^ signaling pathways, dopamine D1 receptor stimulation can modulate the expression and phosphorylation of DARPP-32 (Dopamine and cAMP-regulated phosphoprotein), which is another important regulator for CREB phosphorylation. Phosphorylation of DARPP-32 can happen at Thr34 or Thr75. These different phosphorylation sites lead to either induction or reduction of Ser133 CREB phosphorylation [19,20]. Neither 45 min nor 6 h after relief conditioning, the level of total DARPP-32 and pDARPP-32 (at Thr34) in the NAC were changed (Figure 5c,f).

#### 3.2.4. 14-3-3 and Neuroligin2

Motivated by the data of a comprehensive collaborative study of our group on the proteome after instrumental learning [21], we also investigated the potential involvement of 14-3-3 and neuroligin2 in relief learning. In all five brain regions investigated here, we found that 14-3-3 expression was not affected 45 min after relief conditioning. However, 6 h after relief conditioning the protein level of 14-3-3 was significantly increased in the NAC (*p* = 0.02; Figure 6, Appendix A). Regarding the expression of neuroligin2, no significant changes in any regions of the study and any timepoints following relief conditions were observed (Appendix A).

## 4. Discussion

The aim of this study was to reveal molecular underpinnings of relief learning in male rats. Based on previous findings, the coincidence activation of NMDA and dopamine D1 receptors in NAC is crucial for relief memory formation [7,8]. Therefore, we focused on signaling pathways underlying NMDA and dopamine D1 receptor activation and analyzed the expression and phosphorylation levels of key proteins within these pathways upon relief conditioning. To do so, rats were either control or relief conditioned. A further group stayed naive, i.e., they were just kept in the home cage and had no experimental experience. During relief conditioning, the offset of the US and the onset of CS was temporally associated (a fixed inter-stimulus interval of 2.5 s) while during control conditioning the presentation of US and CS was randomly distributed, i.e., CS and US was mostly unpaired but could also be forward or backward paired [4]. 45 min and 6 h following the conditioning procedures, brain dissection was performed. Of note, an additional group of rats was tested one day later for retention of conditioned relief. In these rats, we observed a robust attenuation of the startle magnitude by the light CS after relief conditioning while the CS had no effects in control-conditioned rats (see Figure 2). Both, the startle attenuation after relief conditioning and the absence of CS effects on startle after control conditioning is in line with previous studies, e.g., [4,5,7,8]. Of note, our control conditioning procedure, i.e., random pairing of US and CS, is considered as the optimal control condition for associative learning [22].

Our molecular analyses were focused on brain regions that have been shown to be involved in relief learning or similar associative learning processes, namely the NAC, PFC, AMY, or HIP while the STR is most probably not involved in such learning processes and, hence, served as an arbitrary control brain region. First, we analyzed the expression and phosphorylation level of CREB. CREB plays a key role in memory acquisition and consolidation [23] and CREB activity within the NAC is critical for the gating of emotional stimuli to animal behavioral response [24]. CREB activity is dependent on its phosphorylation status that can be modulated by various upstream protein kinases or phosphatases [25]. CREB phosphorylation can occur at three different sites in which phosphorylation at Ser133 induces the transcription and translation of CREB target genes such as immediate early genes [26]. Although the critical role of CREB in memory formation is very well studied, there is limited information about the modulation of CREB expression and phosphorylation upon learning. Of note, long term memory formation is dependent on one or two waves of protein synthesis [27,28]. For example, in a one-trial context-dependent fear conditioning, two phases of CREB phosphorylation were determined, the first phase was within 0–30 min (highest level after 7 min) and the late phase was between 3 and 6 h after conditioning [14]. In another study of fear conditioning, it was demonstrated that the paired association of foot shock (US) and tone (CS) leads to a monophasic pattern of CREB activation 15 min after conditioning. While in an unpaired paradigm, a biphasic pattern (within 1 h and 9–12 after conditioning) was observed in the hippocampus (CA1) [28]. Therefore, the present study was designed to include two time points. Since the first phase of activation is mainly expected to happen within 1 h after conditioning, we selected the time point of 45 min after conditioning to evaluate a potential first phase of CREB activation. A further analysis was then performed 6 h after conditioning to evaluate the second phase of CREB activation. 

To demonstrate the involvement of CREB, the expression of CREB and phosphorylated CREB-Ser-133 were studied in all candidate regions of the brain at both time points. Western blot data revealed a biphasic pattern of CREB phosphorylation in the NAC. First, 45 min after conditioning, there was a decrease of CREB phosphorylation, and later, 6 h after conditioning, the CREB phosphorylation level in relief-conditioned animals was significantly higher than in the control group. Both processes may be caused by the activation of NMDA receptors during relief conditioning. NMDA receptors are linked to protein phosphatases and kinases, and activation of NMDA receptors in extrasynaptic or synaptic regions of neurons lead to pCREB dephosphorylation or CREB-phosphorylation, respectively. Calcium-dependent signals per se can also act as a switch between intracellular signaling pathways that can modulate neuronal plasticity through CREB phosphorylation and/or dephosphorylation in different types of learning [29,30]. Our findings point towards a dual role of the NMDA receptor in CREB phosphorylation and dephosphorylation in order to modulate plasticity.

Interestingly, the decreased CREB phosphorylation in the NAC at 45 min was accompanied by decreased CREB phosphorylation in the hippocampus in relief-conditioned rats (Figure 3a,b). However, in both areas, the pCREB/CREB ratio was not affected. In contrast, after 6 h, when increased phosphorylation ratio of CREB in the NAC was accompanied by decreased CREB phosphorylation within the PFC, pCREB/CREB ratio was also significantly changed. The PFC is involved in the inhibitory control of behavioral and cognitive processes [31]. Hence, decreased CREB phosphorylation in the PFC may reflect decreased inhibition of the NAC by the PFC during relief learning. 

After analyzing the modulation level of CREB, we further asked which signaling pathways are involved in the observed CREB phosphorylation in the nucleus accumbens during relief learning. Several studies demonstrated that CREB phosphorylation at Ser-133 may occur through any of its upstream protein kinases including protein kinase A (PKA), Ca^2+^-calmodulin-dependent kinases (CaMK), protein kinase B (PKB or Akt) extracellular-Signal-regulated kinase (ERK) or mitogen-activated protein kinase (MAPK) protein kinase C (PKC), p90 ribosomal S6 kinase (p90RSK), casein kinase I, and casein kinase II [12,13,32]. Based on previous studies, NMDAR induction plays a critical role in relief acquisition [9,10] and is able to stimulate CREB activity through different pathways [17,18,33]. One of the candidate kinases downstream of NMDAR is ERK that can phosphorylate CREB at Ser-133 and is activated upon phosphorylation by MAPK kinase (MEK) [17]. Activation of MEK is regulated by the activity of its upstream kinase, MAP2K. In addition to ERK, CREB phosphorylation also occurs by calcium influx signal, mediated by the Ca^2+^-calmodulin-dependent kinase (CaMKII) [18]. Hence, the expression levels of CaMKIIα, MAP2K, and ERK1/2, also the phosphorylation ratio of ERK1/2 were analyzed in all three experimental groups at both time points. The data from all five regions of the study showed no different expression levels of these candidate proteins neither 45 min nor 6 h after conditioning. However, there were non-significant increases of p ERK1/2 and CaMKIIα in the NAC 6 h after relief conditioning which became significant after analyzing the two proteins together. This may indicate that ERK1/2 and CaMKIIα in NAC play a role in CREB phosphorylation during relief memory formation, but probably the support from other pathways or kinases are also required in order to regulate CREB activity.

In addition to NMDA receptors, stimulation of dopamine D1 receptors is also required for relief learning [10]. PKA is one of the downstream targets of dopamine receptors that can phosphorylate CREB at Ser-133. It is reported that cAMP, by activating PKA, can regulate memory formation and plays a pivotal role in specific types of long-term synaptic plasticity. Moreover, PKA can phosphorylate various downstream kinases and transcription factors that are necessary for memory formation and many other biological processes [34]. In this study, the expression of PKA and its phosphorylation at Thr-197 was analyzed. As shown in Figure 5a, the phosphorylation level of PKA was significantly increased in NAC 6 h after relief conditioning. This suggests that in relief learning, PKA is one of the main kinases that phosphorylate CREB at Ser-133 or maintains the phosphorylated level of CREB during learning. Dopamine receptors can also modulate the activity of Akt. Akt activation is mediated by PI3-kinase and occurs via phosphorylation at two different residues, Thr308 and Ser473. Following phosphorylation, Akt modulates the expression and phosphorylation of CREB at Ser-133. Based on the findings of Brami-Cherrier (2002), either D1 or D2 agonists induces phosphorylation of Akt solely on Thr308 [35]. Therefore, in this study, the expression of total Akt and its phosphorylation ratio on Thr308 was studied in the NAC. In Western blot, no signal of pAkt after 45 min was observed and after 6 h, there were no differences in expression and phosphorylation of Akt between the two groups (Figure 5b,e). Similar results were observed with DARPP-32 expression and phosphorylation (Figure 5c). DARPP-32 is another important regulator for CREB phosphorylation. PKA and Ca^2+^ signaling pathways through dopamine D1 stimulation mediate the expression and phosphorylation of DARPP-32. The activation of DARPP-32 is dependent on its phosphorylation state and modulated by different kinases and phosphatases. DARPP-32 has four sites for phosphorylation: Thr-34, Ser-75, Ser-97, and Ser-130. It is reported that, following the induction of dopamine D1 receptor, PKA can phosphorylate DARPP-32 at Thr-34 and simultaneously, dephosphorylate Thr-75 and Ser-97 by PP2A/B56δ, resulting in amplification of the dopamine D1 signaling and inhibition of protein phosphatase1 (PP1) [36]. Phosphorylation at Thr34, in turn, induces the phosphorylation of CREB at Ser-133 while, phosphorylation at Thr75 leads to a reduction of Ser-133 CREB phosphorylation [19,20]. Moreover, the activation of NMDA or AMPA receptors through protein phosphatase2B (Calcineurin) leads to the dephosphorylation of DARPP-32 at all 4 sites. Therefore, when DARPP-32 is dephosphorylated at Thr-34, it cannot be able to inhibit the activity of PP1 [36,37]. Since the phosphorylation level of PKA had increased after relief conditioning in NAC, we expected to have a higher amount of phosphorylated DARPP-32 at Thr-34, but as it is depicted in Figure 5c, phosphorylated Thr-34 in both conditioned groups was decreased compared to the naive group. We, thus, hypothesized that—at least 6 h after relief learning—the phosphorylation of Thr-34 can be regulated by both dopamine and NMDA receptors at the same time. These data are in concordance to previous findings that coincident activation of dopaminD1 and NMDA receptor in NAC plays a key role in relief learning [10]. 

In addition to the signaling pathways underlying dopamine D1 and NMDA receptors, other proteins may also be involved in relief memory formation. A recent comprehensive analysis on protein rearrangements in mouse brain after auditory learning revealed several protein candidates, including 14-3-3 and neuroligin2, to be overexpressed upon learning [21]. 14-3-3 is a group of regulatory proteins mainly expressed in brain, particularly at synapses. Of note, 14-3-3 is localized in glutamatergic synapses and regulates NMDA receptors [38]. Previous studies demonstrated a positive regulatory role of 14-3-3 in associative learning, memory, and synaptic plasticity [38,39,40]. The 14-3-3 protein family is also part of the Raf-MEK-ERK pathway and can phosphorylate CREB at Ser-133 [40,41,42,43]. Based on the results, we can conclude that in NAC, 6 h after relief conditioning, 14-3-3 may through MEK-ERK signaling contribute to CREB phosphorylation at Ser-133 (Figure 6b). Another candidate was neuroligin2 (NLGN2), which belongs to the neuroligin family including five types of synaptic cell adhesion proteins (NLGN1, NLGN2, NLGN3, NLGN4, NLGN4Y). Neuroligins are located in the postsynaptic membrane and bind to neurexin, their counterparts in presynaptic membrane, and play important roles in synapse formation [44]. In this study, no effect of relief conditioning on NLGN2 expression in any region and at any time point was observed. 

There are several limitations of our study. For example, we only tested male rats and sex might affect the results. However, we believe that sex is not a critical factor in relief learning since we did not detect sex differences in relief learning in previous experiments (Mohammadi & Fendt, unpublished data). Another limitation is that CREB can also be phosphorylated by cGMP-PKG [45]. However, this pathway seems to be more relevant for early memory consolidation while cAMP/PKA signaling is more important for late memory consolidation [45]. Since we were mainly interested in long-term memory, we here focused our molecular analyses on cAMP-associated signaling pathways. However, we cannot exclude the involvement of cGMP-PKG in relief learning. In addition, we can also not exclude that other brain structures than those which were analyzed in this study may be involved in relief learning. One of these further brain structures, for example, could be the bed nucleus of the stria terminalis which plays a crucial role in a behavioral protocol of “ambiguous” threat conditioning [46] which is very similar to relief conditioning. A further limitation is the rather poor spatial resolution of protein expression patterns by using brain dissections for sample collecting. Immunohistochemical staining would have a better spatial resolution and would also allow analyses on a subnucleus level, e.g., shell vs. core region of the nucleus accumbens. However, using Western blot enabled us to measure the expression and phosphorylation of numerous proteins in one sample which is not possible with immunohistochemistry.

## 5. Conclusions

Previous studies of our group, in rodents revealed a critical role of coincident activation of dopamine D1 and NMDA receptors within the NAC in relief learning. The aim of the present study was to decipher the molecular pathways involved in relief learning, by focusing on the signaling pathways underlying NMDA and dopamine D1 receptors. We found robust CREB phosphorylation in the NAC, 6 h upon relief learning (Figure 7). Lu et al. reported that phosphorylation of CREB can be realized via cAMP and/or cGMP, and this dual activation may enhance the activity of CREB [47]. Our data further indicate that CREB phosphorylation during relief learning was mainly induced by PKA phosphorylation, induced by dopamine D1 receptor stimulation, most probably amplified by accompanied weak ERK1/2 phosphorylation and increased CaMKIIα expression, both induced by NMDA receptor stimulation. Therefore, we can assume that in relief learning, phosphorylation of Ser-133 CREB via different kinases and/or independent signaling pathways, in a parallel and/or synergistic way, may magnify the activity of CREB.

Taken together, our study strongly supports a critical role of the NAC in relief learning and is the first one elucidating the involved signaling pathways.

## Figures and Tables

**Figure 1 cells-10-00238-f001:**
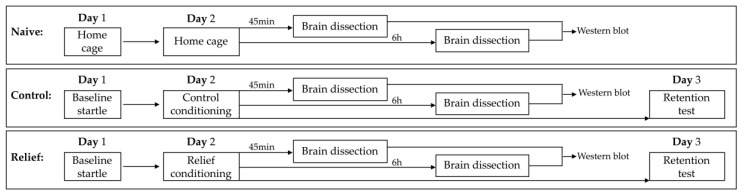
Experimental design overview.

**Figure 2 cells-10-00238-f002:**
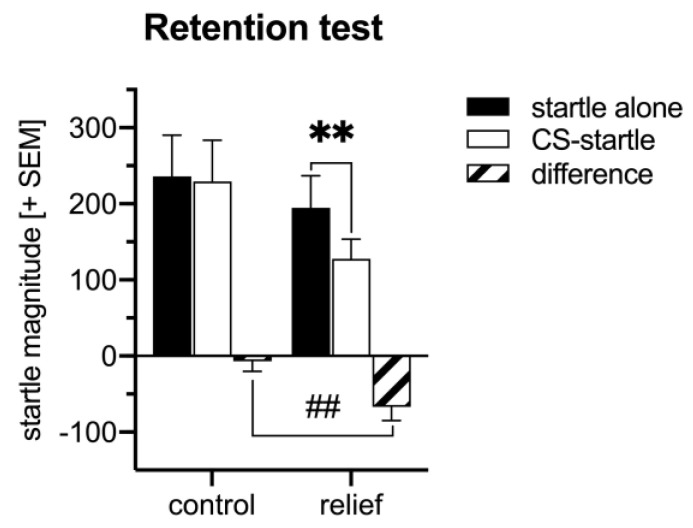
The retention test on conditioned relief showed significant attenuation of the startle magnitude in relief-conditioned rats but not in control-conditioned animals (*n* = 12/group). ** *p* < 0.01, ^##^
*p* < 0.01.

**Figure 3 cells-10-00238-f003:**
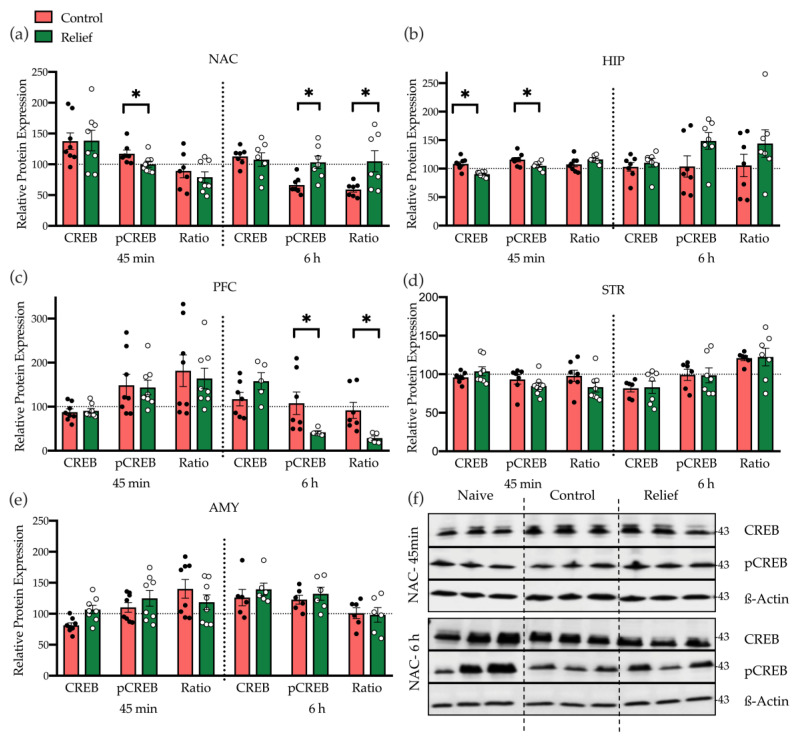
Expression and phosphorylation levels of CREB 45 min and 6 h after control and relief conditioning. (**a**) In the nucleus accumbens (NAC), the phosphorylation level of CREB was lower after 45 min but higher after 6 h in relief conditioned rats than in control-conditioned rats. (**b**) In the HIP, CREB, and pCREB expression were lower 45 min after relief conditioning, compared with control conditioning. (**c**) In the PFC, a significant decrease of pCREB and the phosphorylation ratio were observed 6 h after relief conditioning. (**d**) In STR and, (**e**) the AMY no differences between the two conditioned groups were seen. (**f**) A representative example of a Western blot representing the total and the phosphorylation levels of CREB in the NAC, 45 min and 6 h after conditioning, showing higher level of pCREB in relief conditioning compared with control conditioning (3 different biological samples/group). Bar diagrams depict the mean ± SEM (*n* = 7–8/group; * *p* ≤ 0.05).

**Figure 4 cells-10-00238-f004:**
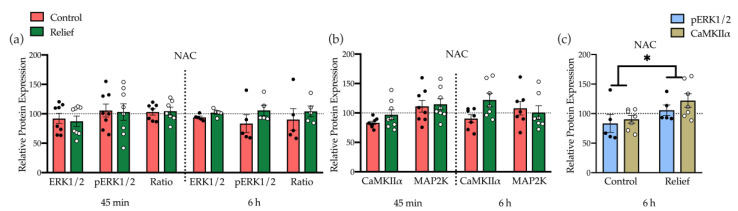
Expression level of ERK1/2, CaMKIIα, and MAP2K in NAC. (**a**) Expression and phosphorylation level of ERK1/2 and (**b**) expression level of CaMKIIα and MAP2K in NAC 45 min, and 6 h after control and relief conditioning. There were no significant differences. (**c**) Analyzing the expression of pERK and CaMKIIα together showed significant differences in the relief-conditioned group. Bar diagrams depict the mean ± SEM (*n* = 5–8/group; * *p* ≤ 0.05).

**Figure 5 cells-10-00238-f005:**
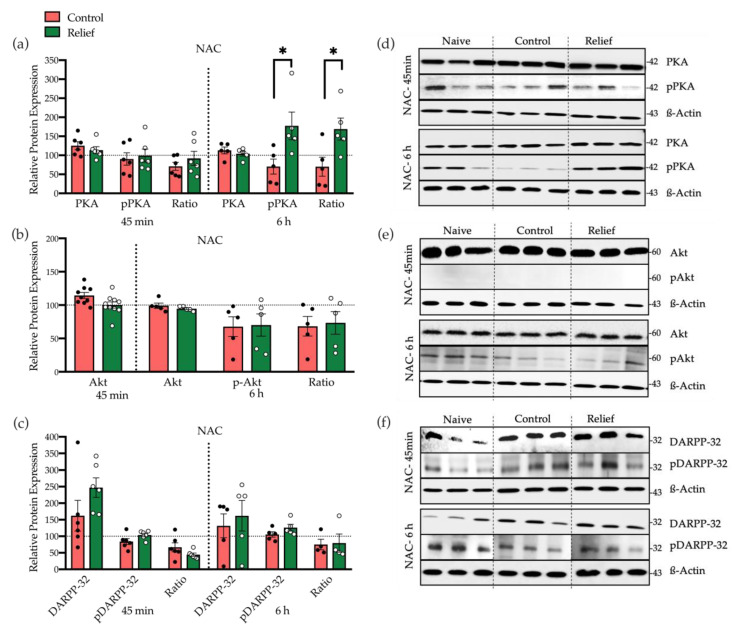
Expression and phosphorylation level of PKA, Akt, and DARPP-32, as well as their phosphorylation ratio in the NAC. (**a**) PKA phosphorylation and the phosphorylation ratio were significantly increased 6 h after relief conditioning in NAC. (**b**) The expression level of Akt was not affected after 45 min and 6 h after relief conditioning. No signal of pAkt was detected 45 min after conditioning while after 6 h the phosphorylation levels of the relief and control groups were not different. (**c**) DARPP-32 expression and phosphorylation levels were not affected by relief conditioning in the relief group compared to the control group. (**d**–**f**) Representative Western blot, showing the expression and phosphorylation levels of PKA, Akt, and DARPP-32, 45 min and 6 h following control or relief conditioning in NAC (3 different biological samples/group). Bar diagrams depict the mean ± SEM (*n* = 5–6/group; * *p* ≤ 0.05).

**Figure 6 cells-10-00238-f006:**
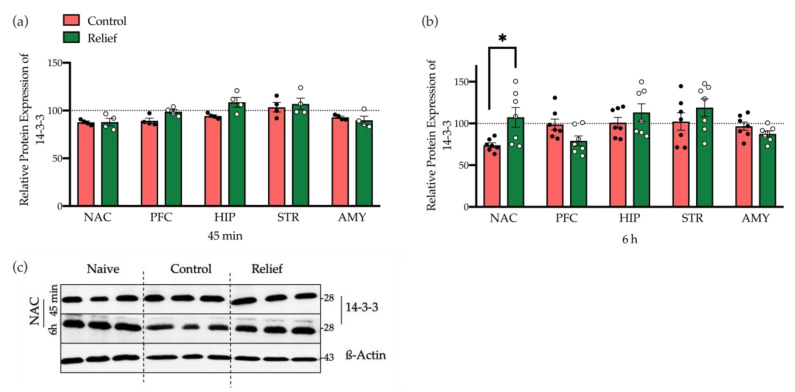
The expression level of 14-3-3 protein. (**a**)The expression level of 14-3-3 was not affected 45 min after relief conditioning in any regions of study. (**b**) 6 h after conditioning the expression level of 14-3-3 in NAC was significantly higher than the control group, while in the other regions no difference was seen. Bar diagrams depict the mean ± SEM (*n* = 4–7/group; * *p* ≤ 0.05). (**c**) Representative Western blot, showing the expression levels of 14-3-3, 45 min and 6 h following control or relief conditioning in NAC (3 different biological samples/group).

**Figure 7 cells-10-00238-f007:**
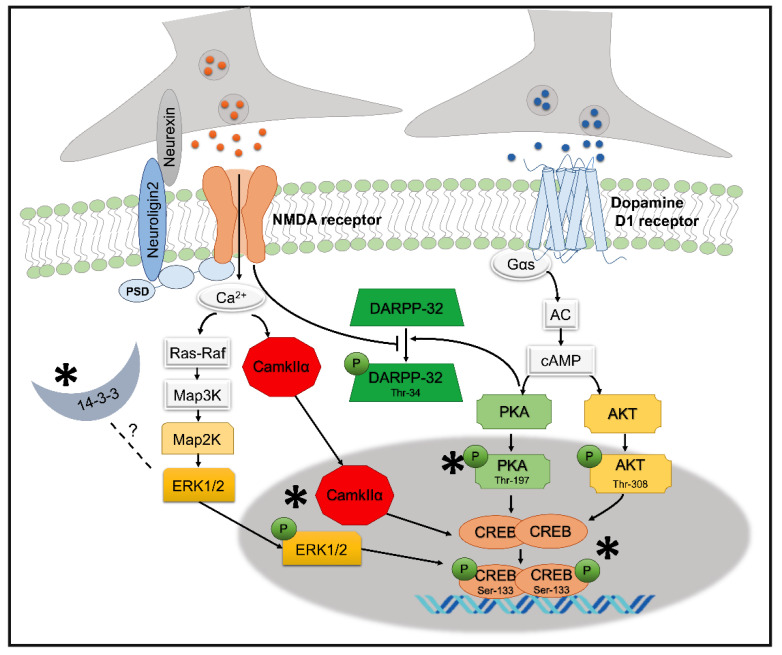
A schematic illustration showing the proposed potential molecular mechanism of relief learning in NAC. Phosphorylation of Ser-133 CREB during relief learning in NAC is induced by a combination of different molecular pathways underpinning dopamine D1 and NMDA receptors.

## Data Availability

Data is contained within the article or Appendix A.

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
