# Peer review of "Regulation of CREB Phosphorylation in Nucleus Accumbens after Relief Conditioning"

_cells, 2021, doi:10.3390/cells10020238_

Round 1

Reviewer 1 Report

SUMMARY:

The knowledge gap this study sought to fill was that little is known about the molecular mechanisms that mediate relief learning in rodents. The authors set up their introduction with references to studies implicating D1 and NMDA receptor activity in relief learning, as well as some supporting the role of CREB in similar associative learning contexts. They further solidify their rationale by referencing the links between D1/NMDA receptors, the various signaling kinases they assessed, and activation of CREB. Overall, they made it very clear why it is reasonable to investigate the particular molecular elements they chose. To do this they first measured the effect their relief conditioning paradigm had on the behavioral response to a startle stimulus, confirming that the paradigm did produce an effect. Following this they used western blot analysis to measure the expression of different signaling kinases. The effect of the paired relief conditioning paradigm was compared to an unpaired conditioning control group and both were measured against a non-conditioned baseline. The difference between phosphorylated and non-phosphorylated forms were measured as well as the ratio between them. Two different time points (45min & 6hr) and multiple brain regions were assessed. The study found, at the 6hr point, a significant increase in NAC expression of pCREB and a decrease in pCREB in the PFC but focused on the NAC where they also found a significant increase in pPKA as well as some minor increases in pERK and CaMKII. These are the main findings from which they draw their conclusions. Those being essentially that relief learning involves D1 as well as NMDA receptor activity which induces phosphorylation of PKA as well as ERK1/2 & CaMKII respectively thereby inducing phosphorylation of CREB which is known to increase CREB activity and contribute to learning and memory.

Critiques:

  • The study essentially performs multiple hypothesis testing on the same samples by using different brain regions and by comparing changes in multiple proteins. As such corrections for multiple hypothesis testing should be performed, although this is not detailed in the methods.
  • It is not described if b-actin controls are run on each blot, or if proteins are normalized to a single separate blot.
  • It is not clear if the Representative western blots are showing a single biological sample in triplicate or 3 different biological samples. Showing Blots of the 45min groups with significant changes between the 6hr groups would improve the figures. Additionally, the blot in figure 5d does not appear to be representative of the data, but rather a more extreme reduction.
  • Sex as a biological variable was not included as a part of this study. The use of male rats, should specifically be added to abstract, introduction and discussion. As sex could influence the results, it is important to note this clearly.

Minor critiques

  • Discussion of limitations of their study and future directions would improve discussion.
  • Would’ve appreciated just a few more sentences framing the impact of the study
    1. What is the scope of the relief learning model?
    2. What will a better understanding of the molecular mechanisms involved help us achieve?

Reviewer 2 Report

It is a nice manuscript, very well conducted, and important to the field. 1. Is the Western-blot the best technique to evaluate cell signaling in brain tissues? What other technique could be used to confirm cell signaling associated with relief learning? 2. What is the difference in relief learning signaling pathways between male and female rats? 3. The authors found CREB phosphorylation in the nucleus accumbens at 6 hours upon relief learning. What about the CREB phosphorylation via cGMP? Is cGMP cell signaling important to consolidate relief learning?

Reviewer 3 Report

Soleimanpour and colleagues analyzed changes in expression and phosphorylation of several signalling molecules in different brain regions after the acquisition of relief memory. I find the experimental design appropriate, utilizing adequate controls and time points, and I enjoyed reading their reflections on the results and discussion.

Having said that, I have some major comments regarding clarity on experimental procedures/data presentation and interpretation of specific results that I hope can help improve the manuscript, as I believe some of the conclusions are not supported by the data. I would recommend for publication if those interpretations are revisited.

Major comments:

1.- The authors should clarify how the ratio of total vs phosphorylated protein is calculated. At first I thought the ratio was calculated within the same protein sample, which I believed is the most adequate procedure, but for example, figure 3a 45min displays n=7 for total CREB in the relief group, but n=8 for pCREB and the ratio. Is the ratio then not calculated from the same protein extraction mix? In a similar way, figure 3C 45 min shows n=8 in CREB and pCREB for the control group, but n=6 for the calculated ratio.

2.- Figure 4. The authors should describe in detail the procedure to “analyze pERK and CaMKIIa expression levels together”. Ideally this should have been performed as an independent experiment. I see no justification to cherry pick pERK as a representative for the NMDA pathway (it misleads statistical power by reducing the number of comparisons).

The possible effect in ERK phosphorylation could be due to the increase in 14-3-3 (Figure S2). What leads to the conclusion that it is the NMDA pathway the responsible of this effect?

I would appreciate images from the WB results included in this figure, as they have rightfully done in all the others.

3.- What happens to pPKA levels in the other areas of the brain assessed in this study? It seems that the effect in pPKA could be the reflection of a very low phosphorylation level in the control group, according to the representative WB images. The intense sensory stimulation without an associative protocol could lead to that effect in a non-specific way, so it would be interesting to show that brain areas not associated with relief learning do not display an increase in pPKA.

Minor comments:

1.- The results displayed in Supplementary figures should be moved from the discussion to the results section

2.- In my opinion the hypothesis of a simultaneous regulation of DARPP32 phosphorylation by dopamine and NMDA receptors requires a better logical explanation. The argument that DARPP32 phosphorylation levels decreased in both control and relief groups loses strength when taking into consideration that a) pPKA levels are only affected in the relief group b) DARPP32 phosphorylation is decreased both at 45min and 6h, whereas pPKA levels are only affected at 6h. How do those differences in training protocol and time fit into the hypothesis sustained by the authors?

Reviewer 4 Report

The article entitled "Regulation of CREB phosphorylation in nucleus accumbens after relief conditioning" by Soleimanpour and colleagues attempt to identity the molecular mechanism underlying the phenomena known as relief conditioning.

The same group has shown before that the activation of both, NMDA and D1 receptors, are required in relief learning. Despite that the process has been described before, identifying the circuits and receptors involved (by this and other groups). The understanding about what signaling pathways leads to the changes in the expression required to present the relief conditioning were elusive. 

The manuscript is well written and concise, the experimental design and the results are clearly described. The experiments presented showed clearly that the activation of NMDA and D1 receptors in NAC lead to CREB phosphorylation via PKA, involving ERK and CaMKII. 

Minor comments/suggestions:

  • Please include in the figures a symbol that helps to identify the significant changes in the band density analyses.
  • It would be helpful to add a "summary" figure to illustrate the potential mechanism proposed by the authors
  • Please add to the supplemental information a figure with all the blots performed in this study
  • Minor comments made on the text

Round 2

Reviewer 1 Report

The authors have addressed my concerns sufficiently and added the some nice discussion on limitations and interpretations.

There are some minor editing needed. For example,

"While there is meanwhile some basic knowledge on the neuroanatomical and 44 neuropharmacological basis of relief learning in mammals [1,7,10] the molecular basis of it has been 45 poorly investigated so far." meanwhile is redundant.

Another minor issues is consistency of Dopamine D1 or D1 dopamine.

Reviewer 3 Report

I appreciate the authors' effort to take my suggestions into consideration. I believe now the conclusions and remarks are fully supported by the data provided.

I understand that the generation of new data is challenging nowadays, considering the world situation, and that is why I am specially appreciative of the authors addressing my concerns regarding PKA phosphorilation in brain areas other than NAc and sharing their results. 

I recommend this article for publication.